# Carbon Recycling of High Value Bioplastics: A Route to a Zero-Waste Future

**DOI:** 10.3390/polym16121621

**Published:** 2024-06-07

**Authors:** Matthew Keith, Martin Koller, Maximilian Lackner

**Affiliations:** 1School of Chemical Engineering, University of Birmingham, Birmingham B15 2TT, UK; m.j.keith@bham.ac.uk; 2Institute of Chemistry, NAWI Graz, University of Graz, 8010 Graz, Austria; martin.koller@uni-graz.at; 3Go!PHA, Oudebrugsteeg 9, 1012 JN Amsterdam, The Netherlands; 4University of Applied Sciences Technikum Wien, Hoechstaedtplatz 6, 1200 Vienna, Austria

**Keywords:** PHA (polyhydroxyalkanoates), externalized costs, plastics, substitutes, alternatives, biodegradation, recycling, waste management, circular economy, biopolymers

## Abstract

Today, 98% of all plastics are fossil-based and non-biodegradable, and globally, only 9% are recycled. Microplastic and nanoplastic pollution is just beginning to be understood. As the global demand for sustainable alternatives to conventional plastics continues to rise, biobased and biodegradable plastics have emerged as a promising solution. This review article delves into the pivotal concept of carbon recycling as a pathway towards achieving a zero-waste future through the production and utilization of high-value bioplastics. The review comprehensively explores the current state of bioplastics (biobased and/or biodegradable materials), emphasizing the importance of carbon-neutral and circular approaches in their lifecycle. Today, bioplastics are chiefly used in low-value applications, such as packaging and single-use items. This article sheds light on value-added applications, like longer-lasting components and products, and demanding properties, for which bioplastics are increasingly being deployed. Based on the waste hierarchy paradigm—reduce, reuse, recycle—different use cases and end-of-life scenarios for materials will be described, including technological options for recycling, from mechanical to chemical methods. A special emphasis on common bioplastics—TPS, PLA, PHAs—as well as a discussion of composites, is provided. While it is acknowledged that the current plastics (waste) crisis stems largely from mismanagement, it needs to be stated that a radical solution must come from the core material side, including the intrinsic properties of the polymers and their formulations. The manner in which the cascaded use of bioplastics, labeling, legislation, recycling technologies, and consumer awareness can contribute to a zero-waste future for plastics is the core topics of this article.

## 1. Introduction

From the invention of Bakelite^TM^ (a formaldehyde/phenol resin) in 1907, to the introduction of Perspex^TM^ (poly(methyl methacrylate)) in 1932, to the accidental discovery of poly(ethylene) (PE) a year later, the plastics industry boomed throughout the 20th century. Hailed as a new wonder material, plastics rapidly replaced more expensive, animal-derived products such as ivory, shell, and leather. Further innovations led to the introduction of a wide range of plastics such as poly(propylene), poly(ethylene terephthalate) (PET), and poly(styrene) (PS). Changes in chemical composition and processing conditions, along with the introduction of additives and cross-linking agents, led to the development of materials with different molecular weights and crystallinities. This, in turn, gave rise to a huge family of materials with highly tunable properties, meaning that plastics found new applications across almost all industrial, commercial, and domestic settings [1]. Due to this versatility, their low cost, and their exceptional durability, plastics have become ubiquitous in our everyday lives, with particular importance in the packaging, healthcare, and textile industries. Estimates vary, but there has unequivocally been a rapid growth in annual production from ~1.5 million tonnes (Mt) in 1950 [2] to between 400 [2] and 460 [3] Mt in 2022. This includes conventional consumer plastic, but also a wide range of industrial engineering plastics, which often go on to form industrial solid waste [4]. Under a “business-as-usual” scenario, this growth rate is expected to continue, with future plastics production forecast to reach ~1300 Mt p.a. by 2060 [3]. Due to their wide-ranging properties, plastics are also likely to play a vital role in a low-carbon future. For example, they can facilitate the lightweighting of vehicles to improve fuel efficiency, reduce food waste through acting as oxygen and moisture barriers in packaging materials, and are an excellent insulator, meaning that they reduce the energy consumption required for the heating of buildings [5]. Indeed, recent research has shown that replacing plastics with alternative materials, such as paper or glass, often resulted in higher greenhouse gas (GHG) emissions [6]. Although other environmental impact factors (human toxicity, ecotoxicity, acidification, eutrophication, etc.) must be considered, it is essential that GHG generation is not inadvertently accelerated by eliminating plastics use from our current materials system, without introducing sustainable substitutes and alternatives.

As the use of plastics will also be fundamental to human activity in future decades, the significant environmental issues they cause must be addressed to avoid additional harm to human and animal health, as well as the wider eco- and atmospheres. Currently, virgin plastics derived from fossil fuels emit an estimated 1800 Mt of CO_2_ annually to the atmosphere, or 3.4% of total global emissions [7]. Macroplastics (those made up of easily visible fragments, typically larger than ~5 mm) may leak into the environment, where they cause significant disruption to ecosystems. Animals are frequently caught in discarded fishing nets, or mistake plastic debris for food; both of these scenarios can lead to death [8]. A more subtle threat comes from microplastics, which are typically less than 5 mm in length [9]. Primary microplastics originate from textile fibers and microbeads, while bulk plastics break down in the environment (through mechanical and other mechanisms), into secondary micro- and nanoplastics. A major source of pollution results from tire wear particles, which have been identified in road dust, soil, waterways, and air [10,11]. Indeed, one estimate has calculated that 5–10% of ocean plastic originated from tires [12], while up to 30% of particulate matter is due to tire degradation [13]. This pollution not only has a significant human health impact, but it is also ingested by a wide range of animals. These bioaccumulate across the food system, with one study showing that microplastics were present in all 32 marine mammal specimens investigated [14]. Here, they can affect feeding, growth, and reproductive systems [15], as well as the finely balanced communication strategies between different organisms [16]. They therefore have a potentially catastrophic effect on the balance of global ecosystems. 

Furthermore, micro- and nanoplastics absorb toxins from the environment, and also introduce the contained additives, of which there are over 16,000, into (human) food. In addition to their toxicity, microplastics also serve as vehicles for the transfer of antibiotic-resistant genes and pathogenic microbes and viruses to different environments [17]. Although the full impact of microplastics on human health is as yet unknown, they have been found in human blood samples, suggesting that they can travel throughout the body and may lodge in organs [18]. In this context, the presence of microplastics in human placentas and human milk was evidenced in 2021 [19] and 2022 [18], respectively. As recently as 2023, it was confirmed that microplastic particles are even able to breach the blood–brain barrier [20]. Additional research has demonstrated that microplastics, at the same concentration in which they are ingested in food, caused death and allergic reactions in human cells [21]. High in the atmosphere, microplastics also have a direct influence on climate change. They undergo astonishing atmospheric transport to the most remote areas, where their occurrence was not expected until recently [22]. Again, the complete effect is unknown, but their presence does impact cloud formation [23] and may accelerate global warming by reducing the albedo of ice and snow [24]. The problem with micro- and nanoplastics is aggravated by their persistence, their sheer volume, and the fact that waste management of these small particles is only recently being discussed [25].

Many of the issues described above stem largely from the widespread mismanagement of plastics waste. To address this requires technical innovation, as well as legislative and social interventions to ensure that the issue of plastic sustainability is tackled in a holistic manner [26]. Historically, the relatively low price of virgin feedstocks, coupled with a lack of recycling infrastructure, led to a throwaway culture (the “take-make-dispose” paradigm), where it was cheaper to landfill end-of-life (EoL) material or manufacturing offcuts than to reprocess them into new products. Beginning in the 1980s, recycling infrastructure was widely developed, with recycled plastic typically costing less than virgin material. However, the price of the virgin feedstocks is intimately linked to the price of crude oil and is therefore subject to volatile price fluctuations. In 2019, for the first time, the price of recycled plastic flakes became more expensive than that of virgin material [27]. The COVID-19 pandemic then compounded the issue beginning in 2020, as the price of crude oil plunged to ~USD 30 per barrel [28], and prices for virgin plastics followed this trend. The introduction of legislation such as the UK’s Plastic Packaging Tax (2022) [29] and the EU Plastic Tax (2022) [30] aimed to mitigate these fluctuations and provide an economic incentive for manufacturers to incorporate recycled material into their products. Both pieces of legislation mandate a minimum recycled content of 30% within plastic packaging. If the recycled content falls below this, then a charge to the manufacturer or importer of GBP 210.82/t applies in the UK (rising to GBP 217.85/t in 2024) [29], while for EU members states, the cost rises to EUR 800/t [30]. It is currently up to member states to then decide how this is paid to the EU; Spain for example, introduced a tax of EUR 450/t (payable by manufacturers/importers) on all non-reusable plastic packaging outside of the medical and agricultural industries [30]. Taxes such as these are useful for encouraging desirable consumer and business behavior. By creating an additional market and increasing the demand for recycled plastics, value is also added to the material, which created an economic driver for recyclers to increase installed capacity. Across Europe, this rose from 11.3 Mt in 2021 to 12.5 Mt in 2022 [31], with forecasts predicting rapid growth in the coming years [32]. Despite this, plastic waste is still likely to be an issue; improvements in technology do not account for human behavior; thus, there will still be leakage into ecosystems, and the generation of micro- and nanoplastics (especially from recycling facilities [33]) will still be a concern. Simply collecting and recycling more plastic waste will not achieve a fully sustainable plastics industry. There is, therefore, a pressing need to address the whole plastics system and replace fossil-based, non-biodegradable plastics with alternative materials which do not result in additional GHG emissions. A non-fossil carbon source and an intrinsically natural biodegradable material is needed to manufacture products according to safe and sustainable-by-design (SSbD) principles [34].

Bio-sourced and biodegradable polymers such as poly(lactic acid) (PLA), poly(hydroxy-alkanoates) (PHAs), and thermoplastic starch (TPS) offer a route to achieving this goal [35]. These materials are a class of polymers with properties such as stiffness, ductility, and elasticity which can be tuned to suit different applications [36,37,38,39,40,41,42]. As such, they have the potential to replace a wide range of conventional polyolefins [43]. This means that they can also be mechanically recycled [44,45,46], thermally degraded [47,48,49,50], and chemically solvolyzed [51,52,53]. In the case of PHAs, chemical recycling approaches, such as pyrolysis, even result in the generation of heavily demanded unsaturated chemical synthons and precursor compounds to be used to produce a range of marketable products [54]. Such biopolymers can also be anaerobically digested and composted, which may be considered a form of biological recycling [55,56,57]. When leakage to the environment inevitably occurs, the materials safely degrade over a short time frame (typically < 1 year) into CO_2_ and H_2_O, although this is strongly dependent on the given environment [58]. The equivalent quantity of carbon is then absorbed as a carbon source during the production of biopolymers: this typically occurs in a fermentation process, in the case of PHAs (polymers produced via fermentation) and PLAs (monomers produced via fermentation), or during the photosynthetic regeneration of starch, in the case of TPS [35]. Recycling the carbon in this way, in addition to mechanical and chemical techniques, means that the widespread adoption of bio-sourced, biodegradable polymers has the potential to drive a sustainable plastics system. The remainder of this article explores the current state-of-the-art of bioplastics, their role within a circular economy, established and emerging recycling technologies which could be applied to bioplastics, and their biodegradation in a range of environments in order to make informed recommendations for a zero-waste future. This workflow is illustrated in Figure 1.

## 2. Current State of Bioplastics

A variety of materials with plastic-like properties are currently referred to as “bioplastics” and are commercialized as such. “Bioplastics” comprise a chemically heterogenous group of polymeric materials with different properties and diverse target areas of application, with novel packaging materials leading the pack. However, there is a need for extreme caution when using this often unclear, contradictory, and misleading expression. As a matter of fact, the International Union of Pure and Applied Chemistry (IUPAC) even advises not to use the term “bioplastics” at all. Indeed, if even scholars with expertise in this field are not able to fully agree on the terminology, it is not surprising that end consumers are confused when confronted with “bioplastics”. Due to this misperception, there is a lack of awareness of the difference between “biobased”, “biodegradable” and “compostable” plastics. It is, therefore, no trivial task for consumers to properly manage “bioplastic” waste [59,60]. For this reason, it is pivotal to globally use legal options to establish generally applicable and binding standards for the classification of “bioplastics”, on which both industry and consumers can rely [58]. The criteria applied to classify a material as “bioplastic” are as follows:(a)Is the material bio-sourced and hence based on renewable resources?(b)Is it biodegradable according to valid norms and certificates, e.g., EN 13432, in relevant environments where established plastics are recalcitrant (soil, freshwater, seawater)?(c)Is it both industrial and home compostable (i.e., can it be put in an organic waste bin or a garden compost heap)?(d)Is it biosynthesized in vivo, or does its synthesis (mostly polymerization of building blocks) require the use of (precarious) chemicals, solvents, catalysts, harsh processing conditions, etc.?(e)Is it biocompatible, and hence, does it not exert any detrimental effects to living beings and the environment? Eco-toxicity tests according to established protocols answer this question.

In its strictest sense, all these criteria need to be fulfilled for the material to qualify as a “bioplastic”. This holistic definition is used by several experts; e.g., Rahardijan et al. [61] postulate that “bioplastics are plastics developed from biobased renewable resources, in which the end plastic products are biodegradable (compostable) by nature under suitable microbial and/or enzymatic decomposition circumstances (temperature and moisture)” [61]. Importantly, many so-called “bioplastics” only fulfill some or even only one of these criteria. Indeed, conventional plastics may be blended with some form of “bioplastic” to enhance that material’s perceived sustainability. Quantifying the amount of bioplastic is specified in the ISO standards 16620 [62], EN 16640 [63], EN 16785-1 [63], EN 16785-2 [63], and ASTM 6866 [64], with some companies using a biomass approach to showcase the incorporation of a renewable element during feedstock production. The subsequent section attempts to provide a brief overview regarding which materials are commercialized as “bioplastics”, and why close attention should be paid when being confronted with this terminology.

From the above description, the designation “bioplastics”, in its strictest meaning, refers to materials which are both bio-sourced and biodegradable under aerobic and anaerobic conditions [65]. This applies to microbially sourced PHAs, in which their production occurs in the aqueous phase under mild conditions of temperature, pressure, and pH-value, according to the biological optima of their microbial production strains. Chitin, a versatile polymer of N-acetylglucosamine produced by numerous diverse organisms, as well as its composites and follow-up products (especially chitosan), also meets these criteria [66]. TPS, consisting mainly of starch and ecologically benign additives for the processing and fine-tuning of material properties (e.g., sugars, alcohols, fatty acids, bio-nanofillers), besides being biobased, was also demonstrated to be biodegradable [67]. TPS attracts attention as a biodegradable material used for casting, extrusion, and 3D-printing processes [61].

Materials like PLA, perceived as the prototype “green plastic” by the general public, are seen as replacements for polyolefin cups, bottles, to-go containers, packaging, films, and textiles [68], and are often lumped under the common umbrella of “bioplastics”. Here, we need to keep in mind that for PLA, only one step in its production chain is “bio”: the fermentative generation of lactic acid by anaerobic microbes. All subsequent process steps (dimerization of lactic acid to dilactides followed by ring-opening-polymerization to PLA) are chemical processes which depend heavily on the use of typically precarious and expensive catalysts and harsh production conditions [69]. Due to its high crystallinity and glass transition temperature, it often also requires industrial composting conditions to biodegrade within reasonable time frames, as highlighted in Section 5.6. Other readily biodegradable and biocompatible polymers like poly(ε-caprolactone) (PCL) are entirely based on fossil feedstocks but are still often labeled as “bioplastics”.

The most problematic group in the context of defining “bioplastics” are the so-called “drop-in bioplastics”, such as biobased PE (bio-PE), biobased PET (bio-PET), biobased PP (bio-PP), and biobased polyamides (bio-nylon). These materials are chemically indistinguishable from their fossil counterparts (PE, PP, PET, nylon), but originate from renewable resources. While it is convenient to implement these “drop-in bioplastics” by using existing infrastructure and machinery, they are no more biodegradable than their petrochemical counterparts, and therefore, are sources of formation of recalcitrant microplastics. Another typical characteristic of “drop-in bioplastics” is the depletion of food resources for their production [70].

All these different aspects make it complicated, especially for laypeople, to judge whether a material commercialized as “bio” rightly bears this label, or whether this is simply “greenwashing” by the manufacturer. Even European Bioplastics criteria state that “Bioplastics are biobased, biodegradable, OR both” [71]. However, to consider a material a real “bioplastic”, standardized material studies are needed to inform accepted certification, which then confirms that it is “biobased” or “biodegradable”, “home compostable”, or “industrial compostable” [71].

In 2021, European Bioplastics estimated a global “bioplastics” production (according to the above broad definition) of about 5.9 million tons per year, which accounts for a negligible share of ~1.5% of the total global plastic production [71]. The annual production of various types of virgin plastics manufactured globally is shown in Figure 2. Note that some sources quote global plastic production in excess of 400 Mt [2,3]; however, this includes a recycling rate of ~9% [72], which is not considered here. Due to the availability of some data, particularly regarding bioplastics, the results shown in Figure 2 are accurate for 2021. This quantity also includes entirely, or at least partly biobased, but not naturally degradable materials such as bio-PE, which originates from microbially produced ethanol [73]. Bio-PE does not undergo biodegradation in aquatic or terrestrial environments, and its production depletes a nutritionally relevant feedstock (cane sugar). Besides being a food resource, the increasing utilization of cane sugar for the production of non-food products like bioplastic or biofuels causes environmental problems, such as deforestation of the tropical forest to provide land area for sugar cane monocultures [74]. Despite these shortcomings, the global bio-PE production, which was commercially undertaken by the Brazilian company BRASKEM in 2010, already exceeds 230 kt annually, and a duplication of this quantity until 2026 is expected [68]. Bio-PET, another example of so-called “bioplastics” is commercialized by a famous soft drink producer to make so called “green bottles”. In 2009, the Coca-Cola Company (Atlanta, GA, USA) launched its “PlantBottle”, made of 100% biobased ethylene glycol and fossil-based terephthalic acid. However, a simple stoichiometric calculation reveals that only about 30% of bio-PET’s carbon content (carbon in ethylene glycol) is bio-sourced, while the major fraction (terephthalic acid) is fossil-based. These “green bottles” consisting of bio-PET are advertised to be easy to recycle; however, repeated mechanical recycling favors the formation of unwanted microplastic particles [73].

Today, about 35.8% of all plastic-like materials labeled as “bioplastics” account for such bio-sourced, but not biodegradable, materials such as bio-PE and bio-PET, while about 64.2% of today’s “bioplastics” are comprised of biodegradable materials like PHAs, PLA, poly(butylene adipate terephthalate) (PBAT), poly(butylene succinate) (PBS), TPS and TPS blends, or cellulose films. Poly(vinyl alcohol) (PVA), PLA, and TPS are currently leading the market; for 2021, their share of the worldwide “bioplastic” market is estimated at 15.4%, 14.1%, and 14.1%, respectively. For microbial PHAs, the most important group of intrinsically natural “bioplastics”, the current share of the bioplastic market is below 2%; approximately 100,000 tons of PHAs were produced globally in 2021. Compared with the total plastic production worldwide, PHA only accounts for roughly 0.01% by mass [78]. 

Also among the group of truly biodegradable “bioplastics”, some do not deserve to be labeled with attributes like “green” and “bio”. For example, PBAT, a material serving as a replacement for LD-PE, is readily biodegradable and compostable, but its starting materials (terephthalic acid, adipic acid, 1,4-butanediol) are predominately produced from fossil feedstocks. This is analogous to the aliphatic copolyester PBS, a material broadly used for manufacturing biodegradable packaging, mulch foils, disposable cutlery and dishes, and medical products. PBS is proven to biodegrade in both fresh and marine water, but its starting materials are only partly biobased: while succinic acid is produced biotechnologically by facultatively anaerobic microbes, 1,4-butandiole is fossil-sourced. The same goes for PCL, a biodegradable material of surgical importance; its raw material, ε-caprolactone, is of fossil origin. Hence, so-called “bioplastics” like PBAT, PBS, or PCL do not fit into the patterns of a circular economy. For PLA production, in turn, the generation of the monomer (lactic acid) catalyzed by lactobacilli is the only biological step in the entire process chain, from raw material to the “bioplastic”. Figure 3 illustrates the different types of “bioplastics”, indicating the attributes which makes them “bio”, according to the established TÜV Austria test scheme for “biobased”, “industrial compostable”, and “home compostable” plastics. Similar labels exist for the America region, with the U.S. Department of Agriculture (USDA) biobased label and certifications provided by the International Sustainability and Carbon Certification PLUS (ISCC+) scheme, which relies on the standard ASTM 6866 [79].

To paint a slightly more optimistic picture, it should be mentioned that the situation is expected to fundamentally change very soon. Until 2026, European Bioplastics expects the entire bioplastic production to increase to about 7.6 million tons, more than tripling the rate from 2020 (about 2.1 million tons). It is therefore likely that the bioplastic trend is here to stay, and it will even witness a boost. PBAT, due to its significance for the production of biodegradable films for packaging, catering items, articles for agriculture, and blends with special properties [80], is expected to take the lead, with 30% of this quantity estimated to be produced until 2026; second place is expected to go to PBS (16%), significantly outperforming PLA (10.4%). PHA, a truly biological and circular material, will, according to this calculation, already be among the top five bioplastics in 2026; with approximately 0.5 million tons announced for production by currently leading manufacturers, an estimated market share of 6.4% of all “bioplastics” can be expected [81]. This illustrates that the market volume of PHA is expected to increase five-fold in just five years. Importantly, for 2026, it is also estimated that PHA will be produced at higher quantities than the pseudo-“bioplastic”, “bio-PE”. This visualizes that the “bioplastic” market is expected to become “greener” during the coming years. Although traditionally used in packaging, an often single-use and low value application, biobased and biodegradable plastics can match and even exceed the properties of fossil plastics. This means that value-added applications, as well as several recycling options, are feasible. To facilitate the sustainable growth of bioplastics, it is important to outline how a circular economy can be embedded within the bioplastics industry, as outlined in the following section.

## 3. A Circular Economy for Bioplastics

Currently, ~98–99% of all plastics produced globally are synthesized from fossil resources, and only about 9% are recycled [72]. Due to poor product design, combining multiple plastic types (e.g., a PET bottle, with a PE label and a PP lid, and sometimes, even a PVC sleeve), there are significant challenges regarding sorting and separating EoL products. Similarly, many plastics contain additives such as colorants or plasticizers, which only compound the issue. As such, the 9% which are “recycled” are often downgraded into a plastic blend, with properties inferior to those of their virgin counterparts [82]. Previous work simulating different plastics systems has estimated that replacing fossil-based feedstocks with sugarcane has the potential to reduce GHGs by 25% [35]. However, even in this scenario, it is crucial that circular practices are implemented from the product design stage, all the way through to EoL to minimize the total environmental impact. Therefore, this section explores the role of a circular economy in maximizing the sustainability of a future biobased plastics system.

### 3.1. Bioplastics Waste Hierarchy

As a result of the issues associated with the current plastics system, it is essential to develop a suitable waste management strategy for bio-sourced and biodegradable plastics. This can be informed by conventional waste hierarchies [83,84,85] and developed using the principles of green chemistry [86] and green engineering [87]. A hierarchy for biodegradable plastics is provided in Figure 4. It highlights that the most preferable option, and for which the greatest influence on environmental impact can be determined, is in the product design phase. In turn, this substantiates that plastic production should be in accordance with the SSbD paradigm. This closely links to reducing the consumption of material within a given product, followed by designing it for reuse. Unfortunately, many conventional plastics, particularly those used in packaging, are only intended for a single use. This has led to the generation of ~160 Mt of plastic packaging waste per year, of which ~85% is disposed of in landfills or discarded as mismanaged waste [88]. When a product has reached the end of its useful life, the value of the material can be retained through mechanical or advanced recycling technologies. Where this is not possible, for example, in regards to highly contaminated plastic waste, incinerating the material and using the energy for electricity and heat generation is preferable to anaerobic digestion (AD). AD requires the construction of additional processes, along with the collection, storage, and transport of biogas before it can be used as a fuel. Each of these steps is likely to contribute to a larger environmental impact than incinerating the material immediately. Industrial composting is a viable alternative to AD; however, due to its aerobic nature, the recovery of useful fuels or chemicals is unlikely. Finally, the least preferred disposal route for biodegradable plastics is landfilling, either in a controlled manner, or through mismanaged practices. Here, they likely degrade at a slower rate than in industrial composting facilities, hence composting is preferable to landfilling. Anaerobic digestion in landfills also produces landfill gas (mostly CH_4_), which has a detrimental effect on the climate when not managed. Ultimately, these materials degrade predominantly into CO_2_ and H_2_O [89], so recycling the equivalent quantity of carbon back into the production of new bio-sourced polymers may still be considered a circular economy approach, albeit along a much larger loop. The principles of a circular economy for bio-sourced and biodegradable plastics are discussed in the following section, while the barriers and enablers to achieving a zero-waste future are presented in Section 6.

Although generally preferred over advanced recycling, it should be noted that the mechanical recycling of fossil plastics results in persistent micro- and nanoplastics [33]. Also, multiple processing steps can lead to thermo-oxidative degradation and reactive byproducts [90]. Hence, mechanical recycling cannot be the only solution for plastics waste management.

### 3.2. Principles of a Circular Economy

The current plastics system is almost entirely linear, following a “take-make-dispose” route. This system is largely reliant on extraction of fossil-based resources, production of petrochemical feedstocks, synthesis of polymers, use, and subsequent disposal. Approximately 6900 Mt of plastics have been generated since 1950 [2,3,72], of which 79% has been disposed of in landfills or lost to the environment, 12% has been incinerated, and just 9% has been recycled [72]. This linear approach is highly resource inefficient, causes the generation of large quantities of waste, and is, therefore, not sustainable. An alternative plastics system can be developed through the application of circular economy principles. These concepts extend back to the mid-1960s and were developed along two main strands. The first concerns the circular flow of materials such that “waste” is returned and used as a resource, while the second involves creating the economic conditions for such a flow to occur. In turn, these elements were developed from the concepts of “industrial ecology” and “industrial symbiosis”, which were coined in the 1940s to describe an optimum location of industries to efficiently use resources [91]. Despite the relatively early introduction of these ideas, it was not until the 1990s that a circular economy was fully defined by Pearce and Turner [92]. There were some later publications on the topic [93,94,95]; however, the idea of the circular economy did not gain traction until the mid-2010s. The Ellen MacArthur Foundation is widely credited with popularizing circular economy principles [96] and, in 2015, the European Commission delivered the Circular Economy Package, which evolved into the Circular Economy Action Plan in 2020 [97]. The foundation of a circular economy consists of three main pillars [96]:The elimination of waste and pollution. The production of waste is a direct result of design decisions made during product development. By introducing the design criterion that at the end of a plastic’s first use, its material must reenter the economy, and the product will be designed for reuse, repair, remanufacturing, or recycling.The circulation of products and materials at their highest value. This principle closely relates to the waste hierarchy, in which products should be kept in use for as long as possible, and when this use is exhausted, the material itself should be reused or recycled. To maintain materials at their highest value, the smallest loops within conventional circular economy diagrams should be followed. These are also typically the loops with the lowest energy demand. A circular economy for bio-sourced, biodegradable plastics is provided in Figure 5.Transition from extraction to regeneration. Through their reliance on petrochemical feedstocks, poor design choices, consumer practices, and a lack of recycling infrastructure, conventional plastics degrade local and global environments. By moving to biodegradable, bio-sourced plastics, carbon can be sequestered from the atmosphere, thus reducing overall emissions. By innocuously biodegrading into CO_2_ and H_2_O, further harm to local environments is also avoided, thus allowing nature to regenerate itself from the harm caused by fossil-based plastics.

Figure 5 demonstrates that by designing for reuse and repair (the smallest amount and the lowest energy, yet the highest value loops), products can be circulated within an economy for a greater time period. When it can no longer be repaired, mechanical recycling and depolymerization offer routes to recover material value. These larger loops correspond to a less-preferred option, illustrated by the waste hierarchy in Figure 4. Even if the material is lost to the environment, this may still be thought of as a circular economy for bio-sourced plastics; biodegradation into CO_2_ and water means that the carbon will be recycled into new materials, thus providing a route to zero-waste. However, it should be noted that this is a last resort; material value is maximized by following the smallest loop, and biodegradation should ideally only occur in controlled settings to prevent uncontrolled side effects, such as leakage of contaminants.

Implementation of circular economy principles, informed by a waste hierarchy, into every step of the plastics supply chain can facilitate a more sustainable plastics system. Improved design, and the development of additive manufacturing technologies, can lead to a drastic reduction in consumption, while improving the ease of repair, remanufacturing, and recycling. This will allow for the circulation of bio-sourced, biodegradable plastics at their highest value, for as long as possible. The reduction in reliance on conventional plastics will subsequently allow for the regeneration of local and global environments. However, it is essential that the current plastics waste crisis is not simply replaced by another environmental issue. Although circular economy principles can offer opportunities for improved sustainability, comprehensive life cycle assessments (LCAs) are necessary to ensure that all environmental impacts are considered, and that proposed solutions are not more environmentally harmful than current practices [98]. These considerations are discussed in the following section.

### 3.3. Environmental Considerations

LCAs are a valuable tool for quantifying a broad range of environmental impacts for the manufacture, use, and disposal of different materials and products. Some research specifically comparing biobased, biodegradable plastics to conventional fossil-based plastics demonstrates an environmental benefit for the former material [99,100,101,102]. However, there is often a large dependence on the end-of-life treatment; non-biodegradable plastics can act as a carbon sink, whereas the uncontrolled degradation of PLA or PHAs (i.e., in landfill or the environment) can result in the release of GHGs, thus highlighting the need for controlled waste management [68,103,104]. Despite international standards (for example, ISO 14044 for LCAs [105] and EN 16760 which applies specifically to bioplastics [63]), there are often large differences in environmental impacts reported across the academic literature, even for the same plastics. This is largely due to the high degree of freedom when specifying an approach (e.g., cradle-to-gate, cradle-to-grave, or cradle-to-cradle), the inclusion/exclusion of carbon credits, the feedstock of bioplastic production (e.g., first, second, or third generation biomass), and the country of operation, which has a significant effect on energy provision, transport emissions, and assumed disposal methods. It is also worth noting that there is added difficulty in directly comparing fossil-based to biobased plastics due to differences in the maturity of each technology. While fossil-based manufacturing processes have benefitted from decades of efficiency improvements through process optimization, biobased plastics manufacturing is largely still in its infancy [106]. As an emerging process, it has not yet seen similar improvements in efficiency, nor gained similar economies of scale. These differences have also been highlighted by numerous reviews of LCAs [107,108,109,110], and thus, critically analyzing different research articles is beyond the scope of this work. It is, however, essential to emphasize that the assumption that bio-sourced products are inherently more environmentally beneficial over fossil-based products must be supported on a case-by-case basis by scrutinizing the ecological impacts caused by the production of a material along its entire life cycle [111]. This section, therefore, identifies and discusses key environmental impact categories which must be considered during the transition from linear, fossil-based plastics to a circular, biobased system.

Possibly the most referred to environmental impact category is energy use. It is generally desirable to minimize energy demand; however, from the wide range of literature reviewed, it appears that biobased plastics do not automatically offer a reduction in energy use compared to fossil-based materials. Collected results from 20 cradle-to-gate studies showed that fossil-based plastics required an energy demand of ~15 to ~82 MJ/kg, while that of biobased plastics required ~3 to ~100 MJ/kg [107]. These wide ranges are a cause for concern, and while bio-PET and PHAs do have a low energy demand in some studies, there is still an overlap with fossil-based sources. In this context, a seminal study reported the impact of the location of a production plant on the ecological footprint of different energy sources used for bioplastic production, in this case, PHA biopolyesters. The ecological footprint was calculated via the Sustainable Process Index (SPI). It was shown that the energy mix (fossil, renewable, nuclear, solar, hydropower, wind, biogas, etc.) used in different studied European, American, and Asian countries resulted in a bandwidth from 372,950 to 956,060 m^2^ of ecological footprint per ton of PHA biopolyester produced. This again shows that it is not merely the net energy demand of a given bioplastic that determines the sustainability of its production, but specifically, the energy provision [112]. It should also be considered that the oxygen and moisture barrier properties of biodegradable plastics may not be equivalent to those of conventional, durable plastics, and hence, thicker layers of the biodegradable plastic would be needed for the same performance in packaging applications. For this reason, 1 kg of a biodegradable plastic may not be sufficient to displace 1 kg of fossil-based plastic; thus, comparisons on a mass basis are less suitable than comparing specified functional units. Where the energy (either electricity or heat) is sourced renewably, there is also likely to be a much lower environmental impact than if energy is supplied through the combustion of fossil fuels. For this reason, comparing energy demands between fossil-based and biobased plastics may not be the most useful environmental metric. Instead, it is the relative emission of GHGs, both directly, and due to energy use, which allows greater insight into environmental impacts.

There are a wide range of GHGs which contribute to global warming and climate change. These include CO_2_, methane (CH_4_), and nitrous oxide (N_2_O), the global warming potential (GWP) of which is quantified in kg of CO_2_ equivalent (kg CO_2_ e). This is typically quoted using a time horizon of 100 years (GWP_100_), as per IPCC guidelines [113]. Some studies have claimed a net negative CO_2_ release of biobased plastics due to carbon uptake during the manufacture of the bioplastic [107,114]. However, these studies are problematic for two reasons. Firstly, they are cradle-to-gate analyses and hence, do not consider the plastics’ end-of-life phase. Secondly, the initial carbon uptake, which was ultimately converted into fossil sources, is not (and should not) be considered. Hence, it does not make for a fair comparison between biobased and fossil-derived plastics. When comparing cradle-to-grave LCAs for biobased and biodegradable plastics to conventional plastics, there may be a reduction in GWP_100_. PHB in particular has demonstrated a significant improvement in GHG emissions [110]. Overall, PLA does seem to contribute towards net GHG emissions, and while some research has suggested that they are significantly lower than petrochemical plastics [102,110], there is a high degree of uncertainty surrounding this idea, largely due to end-of-life treatment. Under international standards, PLA is currently marked as a “7”, denoting “Other”, meaning it cannot be recycled and is therefore often disposed of in landfill. Here, it degrades into CO_2_ and CH_4_ and as such, contributes a significant portion of GHGs [104]. In addition, it is important to stress that PLA, while being industrially compostable (only degradable at elevated temperatures over 60 °C), it is not home compostable [115]. This not only emphasizes the need for improved waste management, but also highlights the complexity of LCAs [110]. Whether biobased, biodegradable polymers have a lower GWP_100_ than conventional polymers is largely dependent on system boundaries, production methods, assumed uses, and end-of-life choices. For these reasons, the plastic type contributing more to the total GHG emissions cannot be categorically stated.

Although other environmental impact categories such as acidification, eutrophication, ecotoxicity, particulate formation, and ozone depletion also exhibit reveal a large range of uncertainty across the published literature, there are some plastics which have been highlighted as the most environmentally harmful [107]. Due to the use of agricultural feedstocks, and their subsequent reliance on fertilizers, biobased plastics, particularly, bio-PET, risk contributing more to acidification and eutrophication than their fossil-derived counterparts [110]. Similarly, bio-PET seems to be the highest in regards to ecotoxicity, with estimates calculated at 2 to 2.5 times higher than that of fossil-derived PET. Biodegradable polymers, such as PLA and TPS, show a similar range to that of fossil-derived plastics [107]. The range for the reported particulate matter formation and ozone depletion are generally similar for fossil-derived and biobased plastics and is largely dependent on the choice of the EoL treatment. For example, incineration causes the formation of a greater quantity of particulate matter than landfilling the same material [116].

Other impact categories which should be considered under the Product Environmental Footprint (PEF) method include the use of various resources such as land, water, minerals, and fossil fuels. As expected, fossil-derived plastics are likely to perform worse than biobased plastics for fossil fuel depletion. However, biobased plastics may demand greater consumption of both land and water, although this is largely dependent on the feedstock used. Where arable crops, such as corn or beets, are used as primary (1st generation) feedstocks, there is direct competition with food production, in addition to large land and water requirements. At present, this competition is insignificant, with just 0.02% of global agricultural land used to produce bioplastic precursors [35]. However, it has been estimated that to replace all 170 Mt of plastic packaging with bio-derived plastics would require 54% of the current global corn production and demand 60% more water than the whole of Europe’s current freshwater consumption [117]. This complete switch is unlikely to occur; however, it does highlight the need for complete consideration of multiple environmental impact factors when assessing sustainability improvements for the current plastics system. To counter this, secondary or tertiary biomass can be utilized, rather than relying on primary agriculture. Secondary biomass includes forestry residues, agricultural wastes, or wastewater streams, which are often rich in nutrients and can be used to grow certain microbial strains. In gas fermentation, based on tertiary feedstocks, these bacteria can recycle the carbon to produce lactic acid, or a range of PHAs when using autotrophic (conversion of CO_2_) or methanotrophic (conversion of CH_4_) microbial strains. These high-value chemicals may then be harvested and further processed to produce biobased, biodegradable polymers, without the high land and water use requirements associated with farming [118,119].

This section has discussed the environmental considerations associated with the widespread adoption of biobased plastics to replace fossil-derived plastics. Significant differences in LCA system boundaries and assumptions mean that it is not possible to comprehensively conclude which material has a lower lifetime environmental impact. Biobased plastics may have the potential to reduce GHGs and lower energy demand; however, other environmental impacts may be worse than those of fossil-derived plastics. Due to the ongoing growth in biobased plastics, there is an urgent need to complete more comparative LCAs to fully understand their environmental impacts, hence preventing unintended and undesirable environmental consequences. For holistic sustainability, socio-economic and ethical factors must also be considered alongside environmental impacts, which the following section aims to do.

### 3.4. Socio-Economic Factors

Both social life cycle assessment (S-LCA) and environmental life cycle costing (ELCC) are relatively new tools first introduced by Benoit and Mazjin (2009) [120] and Hunkeler, et al. (2008) [121], respectively. Recently, an international standard for S-LCA has been introduced [122], but only codes of practice exist for ELCC [123]. As such, there is little academic literature on these topics, and those that do exist are often even more difficult to compare than LCAs. Published studies tend to focus on well-established products (i.e., conventional fossil-derived plastics, rather than biobased plastics) due to the availability of data. This section, therefore, aims to provide an overview of both the social and economic factors which must be considered during the widespread adoption of bioplastics, rather than an analysis of whether biobased or fossil-derived plastics are preferable.

S-LCA is a methodology to inform decision makers by analyzing the social impacts of a product or process throughout its life cycle [124]. Specific social issues can be grouped into five stakeholder groups, which are analogous to the environmental impact categories found in LCAs. These are summarized in Table 1 and are adapted from Ref. [120]; however, it should be noted that there is some dispute over these, including how they may be quantified [125]. Further, there is a dearth of literature focusing solely on bioplastics, with most S-LCA work considering a broad range of biobased products such as fuels, pharmaceuticals, cosmetics, and food additives, in addition to polymer precursors [126,127,128,129,130,131]. However, agricultural upstream processes for various bio-chemicals are assumed to have similar social impacts to those of bioplastics, with the country of origin as the most important determinant regarding the magnitude of each impact [109]. Previous work analyzing the manufacture of bioproducts has identified poverty reduction, rural development, and job creation as the primary positive social impacts [129], while the risk of exploitative working conditions, alienation, and a reduction in quality of life due to harmful effluents have previously been identified as the major negative social impacts [128]. Concerns have also been previously highlighted regarding fair working practices, human rights, and worker health and safety, with biobased products performing similarly to their fossil-derived counterparts [127,131]. It should be noted that many of the countries with climates suitable for the cultivation of bioplastics’ raw materials (i.e., those in the global south) are also those with relatively weak legislation and low social standards. It is, therefore, of paramount importance that the major consumers of plastic products (those in the global north) are aware of potentially continuing negative social impacts during the transition from fossil-derived to biobased plastics.

E-LCC was developed as an economic tool to complement LCA studies and considers all financial costs associated with a product’s life cycle. These economic assessments aim to identify where all costs are incurred, their magnitude, and to whom they are allocated. They can, therefore, be used to identify risks and potential opportunities for the adoption of bioplastics, hence supporting policy and product development to accelerate the transition to a bioeconomy [132]. Unfortunately, E-LCC studies suffer from drawbacks similar to those of S-LCA work, with the added difficulty that, for products under development, such as biobased biodegradable plastics, companies are reluctant to publish details on cost-drivers, potential revenue, and economic sustainability due to the risk of losing a commercial advantage [109]. There are numerous published techno-economic assessments [133,134,135,136,137,138]; however, due to a lack of standards and approaches, along with variation in the products analyzed and geographical location, they are not comparable. Some work has estimated that biobased polymers are price-competitive with their fossil-derived counterparts [137,138], which is highly promising, when considering the need for alternative chemical feedstocks. Not surprisingly, where environmental cost burdens are considered, the relative cost–benefit ratio of biobased products has been shown to improve [139]. Furthermore, the economic viability of biobased products is strongly influenced by volatile oil prices; when prices surge, as they did in 2020–2022, biobased plastics become more price competitive. Similarly, the economic viability of bioplastics is heavily dependent on government subsidies, which vary around the world. In turn, these subsidies are defined by national and international policy and legislation, the impact of which is explored in the following section.

### 3.5. Legislation

Globally, biobased and biodegradable plastics are typically not price competitive with those based on established petroleum feedstocks [140]. Many of the UN’s Sustainable Development Goals (SDGs), which were published in 2012 [141], apply directly to the transition to biobased plastics. Notably, these include SDG3 (good health and wellbeing), SDG11 (sustainable cities and communities), SDG12 (responsible production and consumption), and SDG13 (climate change). In this context, the interrelationship of microbial PHAs biopolyesters and individual SDGs was recently discussed [142]. The 1989 Basel Convention on global hazardous waste was also amended in 2019 by 187 nations to include the export of plastic waste [143]. The convention further recommended that individual countries facilitate the economic and ecological competitiveness of bioplastics [144]. More recently, the 3rd session of the Intergovernmental Negotiating Committee (INC-3) saw 175 nations agree to develop a legally binding agreement regarding the tackling of plastic pollution by 2024 [145]. Additional international organizations, such as the World Economic Forum, McKinsey & Company, and the Ellen MacArthur Foundation, are also promoting science-based policies for a circular, biobased plastics industry [146]. However, even with these global initiatives, the transition towards bioplastics and zero-waste remains slow and requires country-specific policies.

The European Green Deal, launched in 2020, provides a framework for EU countries to become carbon neutral by 2050. Within this, as well as within the Circular Economy Action Plan, there are several policies directly targeting plastics. This includes bans on single-use plastics [147], restrictions on low-grade plastic waste exports [148], and a tax on non-recycled plastic currently set at EUR 0.80 per kg [149]. For bioplastics, there is no specific legislation, but a framework, published in November 2022, has specified how biobased and biodegradable plastics can be used to achieve a more sustainable future [150]. This includes the standardization of biodegradation, as discussed in Section 5. It is hoped that successful incorporation of this framework by individual EU countries will enable efficient carbon recycling and thus improve the circularity of the current plastics system.

In the USA, there are several policies to develop and support a bioeconomy; however, historically, the focus has been on the production of biofuels, rather than bioplastics [151]. This focus has shifted in recent years, with the Biden administration announcing “bold goals” in 2023, which aim to replace 90% of current plastics with biobased alternatives [152]. Additional legislation, such as the Break Free from Plastic Pollution Act, currently limits the use of single-use plastics and prevents the export of plastic waste to non-OECD countries [153]. However, financial incentives currently promote fracking and shale gas production, which in turn, makes fossil-derived plastics cheaper than biobased materials [154]. Despite some steps towards facilitating a zero-waste future, there are currently strong contradictions within existing legislation which makes achieving these “bold goals” economically challenging.

Some positive action is also being taken by China, the world’s largest producer of single-use plastics, with a ban on non-recyclable material by 2025 [155]. To achieve this, there is currently a plan to increase the production of PLA, PBAT, and PBS to a total of ~2 Mt per annum; however, there is not yet a defined waste management strategy to cope with this increased volume [156]. As stated previously, the uncontrolled degradation of these materials can lead to the generation of greater quantities of GHGs than those created by their fossil-derived counterparts. Other Asian countries, such as Japan, Malaysia, and South Korea, have implemented financial subsidies for bioplastics [157], thus demonstrating a commitment to transitioning away from fossil resources.

The introduction of various policies, frameworks, and incentives around the world is a promising step towards the adoption of biobased and biodegradable feedstocks. However, along with this, the phasing out of subsidies for fossil resources, alongside transparent waste management strategies, are essential to develop an economically competitive, carbon neutral, and zero-waste plastic system.

### 3.6. Labeling

Effectively recycling carbon and achieving a fully circular economy relies on maximizing material value, hence keeping plastics in use for as long as possible. To this end, effective sorting of different polymer types at the end of their useful life is essential to prevent the downgrading of different materials. This means that each individual product, or a product’s components, should be labeled with that polymer type for identification purposes. Traditionally, this has been done through manual sorting at waste management facilities; however, it is increasingly achieved through automated, electronic systems. This section aims to provide a brief overview of different labeling strategies for bioplastics so that mixed waste streams can be separated and subsequently recycled into high value products.

Typical identification labels on plastic products (such as those defined in ASTM D7611/D7611M-20 [158]) consist of a number from 1 to 7, within a triangle indicating the dominant plastic type. These are restricted to materials which are almost entirely derived from fossil resources, such as PET (labeled as 1), PP (labeled as 5), and PS (labeled as 6) [159]. Unfortunately, there is no distinct number for biodegradable plastics such as PLA which means they are labeled as a 7, indicating “Other”. As a result, they are not usually recycled and are disposed of through incineration or landfill, meaning the value of the material is lost [160]. The EU, UK, and US all use different symbols to indicate whether a product is recyclable or not. These are generally well established, having been in use for a number of years. More recently, labels indicating the biobased content and a product’s compostability have been developed, with standards such as ISO 16620 (international), EN 16640 (EU), and ASTM D6866 (USA) describing the specific criteria to use [161] There is not, at present, a legal requirement for producers to advertise the quantity of bio-derived carbon in their products [71]; however, any such claims must be substantiated by following the information in one these standards.

There is also a need for general consumers to understand the difference between industrial composting (typically achieved at 40–70 °C over a 6-month period) and home composting. The latter variety relies on lower temperatures (20–30 °C) and much longer time frames. The exact testing conditions used to define whether a material can be classified as biodegradable, industrially compostable, or home compostable are provided in the standards EN 13432 and ASTM D6400 [35]. To strive for a fully circular bioplastics economy, clear and transparent labeling must be used on plastic components to ensure that they enter the correct recycling stream. Although circularity starts with good design, manufacturing waste and EoL bioplastics which must be properly managed will inevitably occur. To this end, implementing an effective recycling strategy is therefore a crucial step in achieving a zero-waste future.

## 4. Recycling Technologies

According to established waste hierarchies, recycling is preferable to biodegradation and composting. “Drop-in” bioplastics, such as bio-PE and bio-PET, can be mixed with their fossil-derived counterparts and processed together to deliver a recycled, partially biobased product. However, more novel, and biodegradable, plastics such PLA and PHAs, potentially contaminate recycling streams, which increases the risk of their rejection. Due to the density differences between unfilled PLA (1.25–1.49 g·cm^−3^), unfilled PHAs (1.21–1.26 g·cm^−3^), and polyolefins (<1 g·cm^−3^), air classifiers, or float–sink tanks, may be used to separate biodegradable from non-biodegradable plastics [162]. Advanced sorting techniques, such as near-infrared, X-ray, or UV spectroscopy, can also be used to help separate waste streams [163], with companies such as Greyparrot even developing artificial intelligence-based sorting systems [164]. Unfortunately, where complex multi-materials are used to achieve certain properties, this type of sorting is ineffective. In food packaging, for example, polystyrene (PS) may be used as a printable external layer, but must be combined with PET to provide a moisture–gas barrier [165]. This highlights the need for improved design, which may be achieved through extended producer responsibility (EPR) schemes [166]. Once this design is achieved, plastics may be recycled following a particular recycling route. Typically, these processes can be defined as either mechanical, chemical (which considers both thermal pyrolysis and solvolysis processes), or biological (which relies on the use of enzymes, or even animals) to digest plastics. This section summarizes the principles of each of these recycling methods and draws on specific case studies from industrial practice and academic literature to present the current state-of-the-art of recycling carbon from bioplastics. Recycling technologies for fossil-derived polymers are more established [104]. Therefore, where there is insufficient information for biobased, biodegradable plastic recycling, the equivalent technology for fossil-based plastics recycling is discussed.

### 4.1. Mechanical Recycling

Globally, mechanical recycling is the most established recycling technology, with the vast majority of recycled plastic waste processed using this technique. It is relatively simple, cheap, and has a lower environmental impact than the manufacturing of virgin plastics [167]. In a typical process, waste is collected, sorted by polymer type, washed, dried, and shredded or ground before being extruded. Depending on the waste feedstock and the desired output, these steps may occur in a different order, or for a different number of times. Following extrusion, the plastic filament generated may then be pelletized for use in conventional manufacturing equipment [168]. Primary mechanical recycling refers to a closed-loop recycling technique where production off-cuts are recycled within the same plant, hence generating a high quality recyclate which can be blended with fully virgin material [169]. Secondary recycling is the processing of post-consumer plastics which, due to degradation during product lifetime and incomplete sorting of the feedstock, likely results in a recyclate with worse mechanical properties than its virgin equivalents. This is typical for the recycling of PET, with some research illustrating a ~40% reduction in strength after five recycling loops [90]. As bottle-to-bottle recycling is not practical over the long term, PET is often downcycled from bottles into fibers and textiles, which are subsequently used for clothing. As these textiles are typically not recycled [170], this approach does not fulfil the definition of a circular economy.

Unfortunately, this downcycling is inherent within the mechanical recycling of plastics. Due to the presence of contaminants or additives within plastic feedstock and the heat and shear stresses applied during extrusion, there is typically a subsequent reduction in molecular weight and mechanical performance [171,172]. This is particularly pertinent to the biodegradable plastics PLA and PHAs, which are thermosensitive [44]. PLA is also particularly hygroscopic. This means that the plastics are degraded, not only by heat, but also by the absorption of moisture, which subsequently causes the hydrolysis of the polymer chains [45]. Thorough drying during the pre-processing stages is therefore of paramount importance to the mechanical recycling of these biodegradable plastics. Further processing difficulties are encountered due to the low glass transition temperature (T_g_) of PLA. Above 55–60 °C, the material becomes “sticky”, making it difficult to extrude into new filament [168]. Similarly, reprocessing can more than double the melt flow index [173], which may subsequently hinder the manufacture of secondary products. To mitigate against the decrease in tensile strength due to molecular weight reduction, it is possible to add chain extenders. The result of this is the generation of recycled PLA, with properties more comparable to those of virgin material [174]. Although some chain extenders have been found not to affect biodegradability [173], the influence of additional additives (like dicumyl peroxide) has not been widely considered [175].

There are a wide range of polymers which are classified as PHAs, with the most well-known type being poly(3-hydroxy-butyrate) (P3HB). Previous research has identified that multiple recycling loops of the P3HB homopolymer result in a significant loss in tensile strength; by the third extrusion cycle, a 50% reduction was reported [46]. Fortunately, the crystalline P3HB lattice can be interrupted by incorporating building blocks other than 3-hydroxybutyrate (3HB), such as 3-hydroxyvalerate (3HV). The resulting copolyesters are more durable for mechanical processing, with five extrusion cycles causing only a 7.1% and 8.3% reduction in tensile and flexural strength, respectively, despite a 16.6% decrease in molecular weight [176]. Similar findings with a lower 3HV content have also been reported [177].

Where reuse is not possible, it is generally acknowledged that mechanical recycling is the preferred EoL treatment due to its simplicity, cost, and environmental impact [171]. From an economic perspective, previous work has estimated that a minimum plant capacity of 5–18 kt per annum is needed for a recycling process to be profitable [178]. However, once the polymer becomes low-grade due to extended recycling loops or the presence of contaminants, chemical recycling techniques could be applied to recover virgin quality monomers, as explored in the following sections.

### 4.2. Thermochemical Recycling

Thermochemical recycling, also called pyrolysis, relies on heat to depolymerize plastics. It is usually carried out in an inert atmosphere, such as nitrogen, to prevent the combustion of the hydrocarbons, which would lead to the production of, predominantly, CO_2_ and water. Instead, the polymer chains are thermally cracked to produce a range of gases, liquids, and waxes which can be subsequently separated and incorporated into virgin petrochemical feedstocks [178]. Compared to the requirements of mechanical recycling, there is a reduced need for effective sorting processes, as pyrolysis can handle heterogenous, contaminated, or degraded mixed plastic waste. However, it generally requires a greater energy input, with typical temperatures of 300–700 °C, with the lower end of this range typically relying on the use of catalysts [179]. Good heat and mass transfer of the plastics to the catalysts are often achieved through fluidized bed technology [180], which has the added advantage of further degrading plastics through attrition. In general, pyrolysis is a well-established technology for the recycling of fossil-derived plastics and mixed-plastic waste (of which it can be assumed that the vast majority is fossil-derived). This is evident from the substantial investments in such technologies by companies such as The Dow Chemical Company in the UK and the Netherlands [181], Plastic Energy in Spain [182], and Klean Industries in Japan [183].

Process conditions, techno-economic analyses, and life cycle assessments of plastics pyrolysis have been widely reviewed in the literature, and although the majority of these reviews focus on fossil-derived plastics [180,184,185,186,187,188], there has been recent attention given to biobased and biodegradable materials, such as PLA [47,48,49] and PHAs [49,50,172]. Lab-scale pyrolysis of biobased and biodegradable plastics can be used to recycle the carbon into virgin quality monomers, and even value-added chemicals. In the pyrolysis of PLA at 450 °C, for example, transesterification or free-radical reactions produce meso-lactide, L/D-lactide, 3-hexanone, 2-propenoic acid, acetaldehyde, and acetic acid as the main products. The product distribution depends on the process temperature, heating rate, and the presence of a catalyst [47]. Various catalysts have been applied to the pyrolysis of PLA, such as metal oxides [189], metal hydroxides [190], and organometallics [191]. This facilitates the use of much lower temperatures, in the range of 250–280 °C, with the metal cations also providing control over the optical activity of the lactide products. As an example, MgO supplied at 5 wt.% of the PLA mass resulted in complete degradation, with a 98 wt.% yield of lactide [189]. The generation of lactide, as a raw material of PLA, demonstrates that this technology facilitates the closed loop recycling of PLA waste. In addition to the use of catalysts, microwave-assisted pyrolysis (MAP) has also been investigated as a potential PLA recycling route. This method facilitates a lower energy consumption rate than other heating methods (such as burning natural gas) and provides fast and uniform heating, as well as good process control [192]. However, in contrast to catalyzed pyrolysis, a wide range of organic products are formed, with a typically lower yield of the lactide monomer [193]. To minimize downstream processing, reduce cost and resource consumption, and maximize the value of PLA waste, it is therefore apparent that catalytic pyrolysis is preferable to MAP.

PHAs exhibit various thermal stabilities, depending on the specific type and the fermentation process used to manufacture it. Therefore, this variation influences the temperature required within a reactor for pyrolytic depolymerization [50]. P3HB, for example, experiences thermal decomposition at temperatures ranging from 225–350 °C [50]. The major depolymerization product is usually 2-butenoic acid, but further decomposition into primary degradation products, such as propene, acetaldehyde and CO_2_, has been observed. The thermal degradation is quite complex, with the reaction auto-catalyzed by the generation of 2-butenoic acid, thus highlighting the need for careful process control. Similar to the MAP of PLA, microwave irradiation is likely to form a wide range of oligomers, although it is up to 100 times faster than conventional, non-catalyzed pyrolysis at 190 °C [194]. P3HB is rarely used as a homopolymer, and it is therefore pertinent to consider the recycling of P3HB blends and PHA copolyesters. Poly(3-hydroxybutyrate-*co*-3-hydroxyvalerate), or P(3HB-*co*-3HV), is one such copolyester which, when pyrolyzed, also results in the production of 2-pentenoic acid originating from the 3HV moieties, alongside the 2-butenoic acid from 3HB. Spinning band distillation (40–110 °C, 50 mbar, 5 h) was able to recover 2-butenoic acid at a purity of 98% [195], which may have secondary applications in the adhesives, coatings, textiles, or cosmetics industries [54]. Interestingly, where alkaline–earth metal (Mg, Al, Ca, and Zn) salts have a strong impact on PLA depolymerization, alkaline metal (Li, Cs, Na, and K) salts seem to have a greater influence on the degradation of PHAs [196]. Other work has, however, found that P(3HB-*co*-3HV) can be easily depolymerized at temperatures as low as 160 °C due to the action of CaO or Mg(OH)_2_ catalysts. Again, this reaction formed 2-butenoic acid and 2-pentenoic acid as the major products, with a significant reduction in the concentration of unwanted minor products, compared to those produced by non-catalyzed pyrolysis [54]. Subsequent separation and re-polymerization of these products demonstrates the further potential for the closed-loop recycling of biobased, biodegradable plastics.

### 4.3. Solvolysis

Compared to thermochemical techniques, the solvolysis of plastics for recycling applications is much less developed. The studies reported herein are, therefore, restricted largely to lab-scale experimental work. These investigations have nevertheless demonstrated the proof-of-concept of the depolymerization of a range of biobased, biodegradable plastics. “Solvolysis” is a broad term which refers to the depolymerization of plastics due to the action of a solvent such as water (hydrolysis), an alcohol (alcoholysis), or glycol (glycolysis); with the latter usually applied to the recycling of PET [197,198]. Generally, the process consists of three main steps [168]:The reactant (either solvent or catalyst) diffuses through the bulk fluid to the plastic surface. Depending on the solvent, plastic type, and reaction conditions, there may be further diffusion into the plastic itself.A chemical reaction occurs, which cleaves a polymeric bond.The resultant decomposition products (and possibly, the catalyst) diffuse out of the plastic and away from the plastic surface.

Hydrolysis is the opposite of a condensation reaction and involves the insertion of a water molecule across one of the ester bonds which are present throughout both the PLA and PHA networks. This can be achieved at lower temperatures than those necessary for pyrolysis, although process conditions depend strongly on polymer crystallinity, solvent, pH, and the solubility of the degradation products [47]. For example, a 95% conversion of PLA to lactic acid has been achieved within 2 h at 160 °C [51], yet other work by the same authors has noted that 3 h at 180 °C and 1.5 MPa was required for this conversion [52]. The hydrolysis of both PLA and PHAs is accelerated through either acidic or basic catalysts. In the latter medium, the process is random chain scission, with concentrations of 0.6 M NaOH reducing the degradation time of PLA to 20 min (180 °C, 1.5 MPa) [199]. Similar results have also been reported for P3HB [53]. Interestingly, pre-treatments of P3HB, such as soaking in methanol, have resulted in an accelerated depolymerization time [200]. This may be due to the modification of the crystalline surface and an increase in the amorphous nature. This subsequently allows for a greater penetration of the water molecules and OH^-^ ions. Acidic solutions cause chain-end unzipping of PLA; therefore, the hydrolysis of PLA and P3HB is autocatalytic due to the generation of acidic monomers [171]. 

Research has also shown that ionic liquids, such as 1-butyl-3-methylimidazolium acetate ([Bmim][OAc]), promote PLA hydrolysis, with a conversion of 94% achieved in 2 h at the significantly lower temperature of 130 °C [201]. It is possible to further react P3HB hydrolysis products into a mixture of PP and CO_2_, valuable precursors to synthetic fuels; or other high value chemicals, such as cumene [202]. For a circular economy, the production of fuels (which are ultimately combusted) is not desirable; however, the generation of value-added products demonstrates the potential for upcycling EoL biobased plastics. There is some evidence that hydrolytic recycling offers an environmental benefit compared to the manufacture of virgin material. For example, it has been estimated that lactic acid manufactured from corn feedstocks requires 55 MJ.kg^−1^, while hydrolysis of PLA only uses 14 MJ.kg^−1^ [203]. However, due to the wide range of polymer types and hydrolysis conditions which may be necessary, the environmental performance of this recycling method compared to virgin production and mechanical recycling is somewhat unclear and depends on a broad range of different factors.

Various short chain alcohols can also be applied to the depolymerization of PLA and PHA. Unlike in hydrolysis, the original monomers are not recovered, but high-value chemicals are created. This waste management strategy may therefore be considered to be the upcycling, rather than recycling, of biodegradable plastic waste. The transesterification of PLA using alcohols forms the corresponding alkyl lactate, i.e., the use of methanol forms methyl lactate, while ethanol forms ethyl lactate. Both substances are inherently biodegradable and exhibit low toxicity and thus, are considered green solvents with the potential to replace petrochemicals in paints, agrochemicals, adhesives, and cosmetics [49]. Typically, lower temperatures are required than those necessary for hydrolysis reactions, but catalysts are almost always required. Zn-based complexes have been used to make alkyl lactates from PLA at temperatures in the range of 40–130 °C [204,205]. Some conversion is reported, even at the lowest end of this range, although not surprisingly, the reaction rate drastically increases with increased temperature and catalyst concentration. At the highest temperatures investigated, it was possible to completely depolymerize PLA in less than 15 min [205]. Alternatives to Zn-based catalysts include N-heterocyclic phosphine (NHP) [206] and [Ca(lac)_2_].5H_2_O [207], both of which also generated value-added chemicals from waste PLA. In addition, acidic alcoholysis can be used to depolymerize P3HB; longer chain alcohols result in longer reaction times and different products. For example, methanol produces methyl 3-hydroxy butyrate, while ethanol yields ethyl 3-hydroxy butyrate [208]. Notably, PHA-derived methyl esters of hydroxy acids were shown to adhere to valid norms for biofuels, with properties similar to those of biodiesel [209]. Alkaline salts may also be used to produce soluble degradation products from P3HB, with 3-hydroxy butyrate, 3-methoxybutyrate, and crotonic acid generated with a 0.5 wt.% NaOH methanol system in 20 min at 110 °C [210]. More recently, ionic liquids have been explored as a novel catalyst for the methanolysis of PHB, with a yield of up to 84% of methyl-3-hydroxy butyrate [211]. In the search for more selective reactants, basic imidazolium acetate ionic liquids have also been explored. These can convert P3HB to crotonic acid with a yield of 97% in the absence of any alcohol, as indicated by Ref. [212].

As biobased, biodegradable plastics are typically polyesters, alternative technologies such as glycolysis and aminolysis could be used in their depolymerization. The application of these systems has, thus far, been limited to the recycling of fossil-derived polyesters, typically PET. Here, a glycol or amine molecule is inserted across the ester bond to recover either the original monomers or their value-added compounds [213]. As biobased PET is chemically identical to its fossil-derived counterpart, it is expected that these techniques could also be applied to this material.

### 4.4. Biological Recycling

As biodegradable plastics can be broken down by microbes and enzymes, biological recycling refers to the action of these biological materials on plastics. Note that it is different from biodegradation (explored in the following section), as the key aim of biological recycling is to recover useful carbon compounds, whereas biodegradation will typically only produce CO_2_, H_2_O, and CH_4_, depending on the conditions. Similar to solvolysis, biological recycling often takes place due to the cleavage of ester bonds by specific enzymes. However, unlike solvolysis, biological recycling is generally very slow, with complete degradation achieved over a time scale of days to weeks. Moreover, the kinetics cannot be accelerated by increasing the bioreactor temperature due to deactivation of the enzymes [168]. As such, biological recycling techniques are still in their infancy and are unlikely to be economically viable in the near future.

Protease, lipase, and cutinase have all been investigated for their potential to decompose PLA, with alcalase (a type of protease) identified as being among the most efficient [214]. Other alkaline proteases have been shown to produce large quantities of lactic acid from PLA; however, acidic and neutral proteases can exhibit almost no activity [215]. The bacteria releasing these enzymes subsequently ingest the lactic acid and use it to fuel their own growth until none of the substrate remains [216]. These systems are generally very sensitive to pH-value; thus, when lactic acid uptake is slow and the pH-value drops to less than 6.5, enzymatic activity can plummet [217]. PHAs are used for carbon and energy storage within their synthesis bacteria, and these bacteria also house enzymes capable of breaking down the PHA into its constituent monomers. However, these enzymes are only present inside the bacterial cells, and thus, extra-cellular depolymerases are needed, such as those secreted by *Bdellovibrio bacteriovorus* [212]. Commercial lipases have also been applied for the depolymerization of poly(3-hydroxybutyrate-*co*-4-hydroxybutyrate) (P(3HB-*co*-4HB)), with the authors confirming that this is a feasible process for controlled degradation [218]. Similar work has identified cyclic oligomers as the main decomposition product, which was readily repolymerized by the same enzyme [219]. As such, this biological process may have the potential for future carbon recycling applications, provided that the previously reported cost and scalability issues can be addressed [220].

This section has provided an overview of the technologies which can be applied to the carbon recycling of biobased and biodegradable plastics. To combat the issues of non-degradable plastic waste and the bioaccumulation of microplastics, biodegradable plastics have the potential to replace conventional, durable plastics. The following section aims to investigate the biodegradation of these materials in a range of environments.

## 5. Biodegradation

As previously discussed, conventional plastics consist of xenobiotic polymers that are not found in nature, and there are no known microorganisms or enzymes to break them down in natural environments. Naturally occurring polymers, such as starch, silk, chitin, or cellulose, on the other hand, are readily biodegraded. The degradation rate of natural materials depends on the environmental conditions. For instance, large logs of wood can withstand centuries in dry environments, or even under water in low oxygen conditions.

Conventional plastics exhibit different stabilities against solvents, and when exposed to the open environment, they are attacked by UV light and mechanical stress. UV radiation will lead to chain scission, which reduces mechanical properties, and various forces can lead to attrition. Hence, fossil plastics can break down into small particles—micro- and nanoplastics—and these will be very persistent, requiring on the order of hundreds of years to fully decompose, in contrast to the degradation of biodegradable “bioplastics”. Biodegradability can be defined according to different standards, where EN 13432, originally developed for packaging materials, is the most common [221]. Other frequently used tests assess the compostability under “industrial” and “home” composting conditions. Marine biodegradability, as well as freshwater biodegradability, are of interest. Note that “biodegradability” evaluates the disintegration of the materials and their final “mineralization”, yielding CO_2_ and H_2_O under aerobic, plus CH_4_ under anaerobic, conditions, due to the action of microorganisms in natural environments and does not consider UV-light and mechanically-induced comminution. The biodegradability can be tested for the polymer (e.g., PLA or PHB), the bioplastic composite (which is the formulation (compound) of at least one biopolymer and its additives and fillers), and the final product made from that material. The biodegradability of PLA, PHAs, and TPS, along with their composites, has been widely reviewed elsewhere [222,223,224,225,226,227], and some examples are provided in Table 2 below. The biodegradability of additives can be grossly different from that of the (bio)polymer(s), and in general, the thicker, the more crystalline, and the less wettable the bioplastics article, the slower its biodegradation will proceed. Cross-linked materials are also, in general, less degradable than separate polymer chains (e.g., vulcanized rubber, which has lost the intrinsic biodegradability of latex). The fate of bioplastics articles in the environment depends on the application (e.g., whether littering is likely), or even if an end of life in nature is foreseen. In rivers, lakes, and the ocean, the density of the material will determine whether it will float on the surface or sink. Bioplastics have a density > 1 g.cm^−3^, so they are prone to sinking and being subjected to anaerobic digestion (again depending on particle size). The most common natural enzymes to break down bioplastics are esterase, cutinase, PETase, polyamidase, intra- and extracellular PHA depolymerases, and PLA depolymerase [228]. The engineering of enzymes for faster bioplastics degradation has been suggested [229], as well as incorporating enzymes into bioplastic formulations for controlled degradability. Bacteria, fungi, and algae can degrade bioplastics.

As these comments indicate, the topic of “biodegradation” is a complex one, where clear standards are needed to avoid ambiguity and misleading claims in the marketplace. Below, the main standards for the biodegradation of bioplastics are described. For an extensive review, see, e.g., Lackner et al. (2023) [230] and Folino et al. (2023) [231].

### 5.1. Industrial Composting

In “industrial composting”, also called “commercial composting”, organic waste is converted into compost in an aerobic process, with the target of treating the waste stream and creating a product of value which can be sold, e.g., to gardeners. In contrast to a compost heap in private settings (see below), industrial composting is characterized by intensive manipulation of the biowaste to achieve short residence times. The input materials include garden and kitchen waste, twigs, grass, etc., where a certain C/N ratio and air permeability must be ensured for proper functioning. Foreign objects in the feed material need to be removed, e.g., plastics articles. The idea of using biodegradable plastics (e.g., for compost bags, tea bags or fruit labels) in an industrial composting setting is that they will biodegrade within the processing time, so that the final compost does not show any visible particles of the material.

A certification body, OWS, writes: “90% carbon to CO₂ conversion at a temperature of 58 °C is required in six months”, and “90% disintegration is required in a 12-week test running at a well-defined (high) temperature profile”. For a product to obtain “industrial composting certification”, additional ecotoxicology tests are required [232]. The applicable standards are ISO 16929:2021 and EN 14045. Another certification body, TÜV Austria, writes: “Products that are solely OK compost INDUSTRIAL-certified are those that compost only in industrial composting facilities (at temperatures between 55 to 60 °C), so products that are solely OK compost INDUSTRIAL-certified should not go into the garden compost. Conversely, OK compost HOME refers to products that also compost at lower temperatures, so they can go into the compost heap in your garden at home, hence the title ‘HOME’” [233]. Their reference is EN 13432 [221]. Conforming products can be awarded the European Bioplastics “seedling” logo. For details on industrial composting, see European Bioplastics [234].

### 5.2. Home Composting

“Home composting” or “garden composting” [235] does not reach the same temperatures as industrial composting. While industrial composting is in the thermophilic regime (50–65 °C), “backyard composting” proceeds in the psychrophilic (0–20 °C) to mesophilic (20–45 °C) temperature range. While PLA will meet the time requirements of “industrial composting” (depending on the thickness of the film), it will not pass “home composting” certification. On the other hand, TPS, P3HB, and other types of PHAs are home compostable (depending on the thickness of the studied article). For details on “home composting”, see European Bioplastics [236].

### 5.3. Biodegradation in Terrestrial Environments

Compost is humid and hot, so it will provide the fastest biodegradation rate of all natural environments. In soil, the process will be slower. The exact rate of biodegradation will depend on the water content of the soil, its temperature, the presence of microorganisms, and the type and shape of the bioplastics material. Microorganisms can secrete extracellular enzymes to degrade bioplastic materials, using them as carbon and energy source. The contribution of biodegradable plastics to soil organic carbon stock is discussed in the work of Guliyev et al. [237].

### 5.4. Biodegradation in Biogas Facilities

The standard ISO 14853:2016 [238], entitled “Determination of the Ultimate Anaerobic Biodegradation of Plastic Materials in an Aqueous System”, was developed to study CH_4_ formation from bioplastics under anaerobic conditions, and the material is exposed to sludge for 90 days. Converting bioplastics into biogas can be a means of energy recovery [55]. Energy recovery is also feasible with thermal processes (incineration or gasification), where the latter allows for carbon recovery by gas fermentation. For instance, the production of P3HB from methane and from syngas was demonstrated. P3HB from CO_2_ can be obtained by cyanobacteria or non-phototrophic autotrophs, which can also close the carbon loop. When bioplastics are sent to anaerobic digestion systems to produce biogas, their thickness must not be too great to be compatible with the processing times, a similar practical boundary condition comparable to industrial composting.

### 5.5. Biodegradation in Freshwater

The aquatic environment includes freshwater and seawater. Rivers have been identified to transport huge fractions of micro- and nanoplastics into the oceans. Primary microplastics include, e.g., car tire attrition (vulcanized rubber) and textile fibers (often PET), while secondary microplastics are formed by the fragmentation of larger plastics articles through photodegradation and mechanical forces. The rate of degradation of bioplastics in freshwater depends on several factors, among them temperature, pH-value, and presence of microorganisms. The certification of “OK biodegradable water” is valid for freshwater and does not automatically guarantee marine degradability [239]. For a review of bioplastics-based microparticles in the aquatic environment, see Ribba et al. [240].

### 5.6. Biodegradation in Seawater

Seawater, in general, is a more challenging environment for bioplastic biodegradability than freshwater due to typically lower temperatures and lower concentration of microorganisms. ASTM D6691 [241], “Standard Test Method for Determining Aerobic Biodegradation of Plastic materials in the Marine Environment by a Defined Microbial Consortium or Natural Sea Water Inoculum”, is carried out at 30 °C, and the conditions are not representative for, e.g., deeper regions of the oceans where dissolved oxygen levels are low. As van Rossum (2021) has pointed out in his review on the marine biodegradability of plastics, the oceans often possess nitrogen limitations, resulting in a slow growth and biodegradation rate [242]. However, a very recent study reports the biodegradability of PHA microbeads in a sea depth of 757 m [243]. OWS offers certification for biodegradation in seawater (pelagic zone, based on ASTM D6691), where 90% carbon to CO_2_ conversion must be reached within six months. They also provide certification for the sediment–seawater interface (eulittoral zone, based on ISO 19679 [244] and ISO 18830 [245]), and in sediment (benthic zone, ISO 22404 [246]) [247]. For a review on bioplastics degradation in the marine environment, see Chen (2022) [248].

To illustrate the conditions necessary, a summary of different biodegradation studies for a range of bioderived plastics is presented in Table 2. This shows that PLA, PHAs, PBAT, cellulose acetate, and starch-based materials can all be degraded in industrial composting facilities. However, for home composting, only PHAs and starch-based materials exhibit significant degradation, likely due to the lower temperature, which is significantly below the glass transition temperature (T_g_) of other materials; for example, the T_g_ of PLA is ~60 °C [249]. To mitigate against leakage of plastics into ecosystems, it is essential that these materials harmlessly biodegrade over short time scales. As illustrated in Table 2, various PHAs do degrade relatively quickly in soil, seawater, and freshwater environments, suggesting that they are a promising material for a zero-waste future.

**Table 2 polymers-16-01621-t002:** Summary of biodegradation options for a range of bioderived plastics.

Environment	Plastic	Form	Conditions	Outcome	Reference
Industrial composting	PLA	Rigid film, 80 × 30 × 0.3 mm	KNEER composting system, 54 to 64 °C	~5% degraded at 14 days, ~85% degraded at 70 days	[250]
	PLA	Bottle, 473 mL, 208 × 52 × 0.3 mm	Cow manure, wood shavings, and food waste, 60 °C	~94% degraded at 14 days, ~100% degraded at 28 days	[251]
	PLA	Rigid film, 40 × 30 × 0.3 mm	Leaves (40%), branches (30%), grass (30%), 52–59 °C	Complete decomposition in 70 days	[252]
	PLA	5 × 5 × 0.035 mm film	Mature compost, 52 to 58 °C	~70% CO_2_ evolution at 30 days	[253]
	P3HB	5 × 5 × 0.035 mm film	Mature compost, 52 to 58 °C	>90% CO_2_ evolution at 30 days	[253]
	PBAT	5 × 5 × 0.035 mm film	Mature compost, 52 to 58 °C	~30% CO_2_ evolution at 30 days	[253]
	Thermoplastic cellulose acetate	100 cm^2^ surface area	Biowaste mixture, 45 to 78 °C, mixed and moisturized every 7 days	19% degraded in 3 weeks	[253]
	PLA/a-PHB blend (15 mol% a-PHB)	Rigid film, 40 × 30 × 0.3 mm	Leaves (40%), branches (30%), grass (30%), 52–59 °C	Complete decomposition in 70 days	[252]
	Thermoplastic cellulose acetate with 5 wt.% double hydroxide sorbate filler	100 cm^2^ surface area	Biowaste mixture, 45 to 78 °C, mixed and moisturized every 7 days	18% degraded in 3 weeks	[254]
	Mixture of polyester starch, PLA, and PHA in varying proportions	Various	Industrial composting site, 66.1 to 73.4 °C, turning 1/week	85% degraded in 49 days, 98% degraded in 128 days	[255]
Home composting	PLA	Powder, particle size < 0.3 mm	30 °C, 45 days	Max. 20% degraded	[249]
	PLA	20 × 20 mm,	10% mature compost, 54% vegetable residue, 36% wood chip, 20 to 30 °C	No significant change after 78 weeks	[256]
	P(3HB-*co*-3HV) (20% 3HV)	Dog-bone shaped tensile test pieces	Home compost, ~20 °C	~70% degraded at 150 days	[257]
	PLA/aPHA blend at 9:1, 7:3, and 5:5 by weight	Powder, particle size < 0.3 mm	30 °C, 45 days	9:1—30% degraded 7:3—70% degraded 5:5—85% degraded	[249]
	Potato starch	Food tray	8 to 18 °C, Nov to May in UK	Up to 100% in 90 days	[258]
Anaerobic digestion (biogas production)	PLA	Powder	Wastewater sludge feed, 38 °C	~80% degraded in 500 days	[56]
	PLA	Powder, particle size 0.5 to 1.5 mm	Wastewater sludge feed, 38 °C	4% degraded in 60 days	[259]
	TPS	Powder	Wastewater sludge feed, 38 °C	~85% degraded in 30 days	[56]
	TPS	Powder, particle size 0.5 to 1.5 mm	Wastewater sludge feed, 38 °C	48% degraded in 60 days	[259]
	P3HB	Powder	Wastewater sludge feed, 38 °C	~80% degraded in 30 days	[56]
	P(3HB-*co*-4HB)	Powder, particle size 0.5 to 1.5 mm	Wastewater sludge feed, 38 °C	73% degraded in 60 days	[259]
	Thermoplastic cellulose acetate	100 cm^2^ surface area	38–40 °C, anaerobic digestion	37% degraded in 2 weeks	[254]
	Thermoplastic cellulose acetate with 5 wt.% double hydroxide sorbate filler	100 cm^2^ surface area	38–40 °C, anaerobic digestion	50% degraded in 2 weeks	[254]
Terrestrial biodegradation	PLA	Powder	25 °C, 60% moisture	14% degraded in 4 weeks	[260]
	PLA—Sisal Fiber composite	Rigid film, 30 × 30 × 1 mm	Alluvial-type soil, 30% moisture	10% degraded in 98 days	[261]
	P3HB	Film discs, 30 mm diameter, 0.035 to 0.045 mm thick	21 and 28 °C, 50% moisture	After 5 weeks, 60% degraded at 21 °C, 95% degraded at 28 °C	[262]
	P3HB	Film	19 to 27 °C, 33% moisture	82% degraded after 80 days	[263]
	P(3HB-*co*-3HV) (12% 3HV)	Film discs, 30 mm diameter, 0.035 to 0.045 mm thick	21 and 28 °C, 50% moisture	After 5 weeks, 90% degraded at 21 °C, 100% degraded at 28 °C	[262]
	P(3HB-*co*-3HV) (8% 3HV)	10 × 10 × 0.2 mm film	11 to 30 °C. 17 to 23% moisture	60% degraded in 112 days	[264]
	TPS	10 × 10 × 0.5 mm film	43% top soil, 43% farm soil, 14% sand, 18 to 22 °C	23.2 to 26% degraded after 280 days	[265]
Freshwater biodegradation	PLA	Rigid film, 50 × 10 × 0.3 mm	69.5 to 70.5 °C	~90% degraded at 3 days, ~100% degraded at 7 days	[250]
	PLA	Rigid film, 20 × 30 mm	Freshwater river and freshwater link	~5% degraded after 1 year in both environments	[266]
	PHA (Danimer Scientific’s Nodax PHA; P(3-hydroxybutyrate-*co*-3-hydroxyhexanoate))	Powder	With activated sludge, 25 °C	~85% degraded in 50 days	[267]
	P3HB	Film disc	Freshwater lake, 19 °C	~96% degraded in 49 days	[268]
	PBAT	Rigid film, 20 × 30 mm	Freshwater river and freshwater link	River—~95% degraded, lake—~23% degraded after 1 year	[266]
	PBAT/PLA	Powder	With activated sludge, 25 °C	~40% degraded in 250 days	[267]
Seawater biodegradation	PLA	Flexible bag and rigid bottle, both pulverized	25 °C	Bag—4.5% degraded, bottle—3.1% degraded, both after 180 days	[269]
	PLA	Film, 50 mm diameter, 0.2 mm thickness	14 to 22 °C	<1% degraded after 260 days	[270]
	PHA (P(3HB-*co*-3HV); Mirel 2200 and 4100)	Film	25 °C	45.1% degraded after 180 days	[269]
	P3HB	Film	30 °C	90% degraded after 100 days	[271]
	P(3HB-*co*-3HV) (8% 3HV)	Powder	25 °C	90% degraded in 210 days	[272]
	P(3HB-*co*-3HV) (12% 3HV)	Film	17 to 20 °C	60% degraded in 42 days	[273]

## 6. Future Perspectives and Challenges

To achieve a zero-waste future within the plastics industry, it is essential to transition from the current linear system to one based on renewable raw materials and underpinned by circular economy principles. Due to their tunability and range of properties, biobased and biodegradable plastics have the potential to replace many conventional consumer plastics. However, the major barrier to their widespread adoption is the current high cost of alternatives (especially PHAs) along with improved public knowledge regarding how to properly dispose of these materials. By designing materials with inherent biodegradability, the issue of microplastic generation during mechanical recycling can be effectively mitigated, while the use of bioderived feedstock can address the sustainability concerns and high energy costs of using fossil resources. Overproduction of materials at the manufacturer level and overconsumption at the consumer level are also fundamental considerations; legislation to tackle the former, with public education to consider the latter, are key methods of tackling these two issues. Finally, there is a concern among producers regarding the compatibility of future materials with existing manufacturing equipment. To counter this fear, there is a need for further research into the application of green chemistry and engineering principles to existing processes. These major barriers and enablers are summarized in Figure 6.

## 7. Conclusions

For the carbon recycling of bioplastics, it is first essential to establish clarity regarding what it means to classify a material as a “bioplastic”. From a holistic perspective, it must be biobased, biodegradable in a range of environments, according to the relevant standards, biosynthesized (i.e., it must not rely on fossil-derived chemicals), and biocompatible, so that it does not exert detrimental effects on existing ecosystems. Even if all these criteria are met, the subsequent bioplastics are not inherently sustainable. To assess this, multiple environmental impacts, such as resource (energy, land, water, etc.) consumption, climate change potential, ecotoxicity, acidification, and eutrophication, must be considered along each stage of a product’s life cycle. In addition, socio-economic and ethical factors must also be taken into consideration. By applying principles of green chemistry and engineering, a waste hierarchy for bioplastics has been developed which, in accordance with the circular economy paradigm, highlights waste prevention and a reduction in resource consumption as the preferred pathways. There is a strong need to further design new plastic materials for reuse, and recycling. To achieve this, relevant legislation has been introduced at a national, or even regional level, as is the case with the EU’s Circular Economy Action Plan. However, many global acts remain limited; frameworks and policies are often not legally binding, and there is still strong support for fossil-derived chemicals, with financial subsidies widely available. Further hindering the transition to a circular economy is a general lack of labeling of different plastic types which leads to significant public confusion regarding whether or not a plastic can be recycled, composted, or must be disposed of in a landfill or incinerator. Biobased and biodegradable plastics such as PLA and PHAs are currently marked with a “7”, meaning “other”. They are hence frequently not recycled due to the potential contamination of other plastic waste streams.

There are, however, a wide range of technologies available which can recycle these materials. Mechanical techniques are the most mature, cheapest, and likely require the lowest energy input. However, the material is at risk of being “downgraded” due to the thermal instability of materials such as PHAs and the degradation of the polymer chains. To combat this, advanced chemical recycling, such as pyrolysis or solvolysis, will play a part in a future plastics system. These technologies use heat, solvents, and/or catalysts to recover virgin-quality monomers which may be repolymerized directly or blended with a fully virgin feedstock to create new plastic products. Pyrolysis, in particular, has received some commercial interest, and is likely to be an economically viable waste management option for biobased, biodegradable plastics. Biological recycling, in which the use of enzymes recovers the monomers, remains in its infancy and will be more difficult to commercialize due to the slow kinetics and lack of scalability.

Currently, the primary barrier to the adoption of biobased and biodegradable plastics is their high cost compared to that of fossil-derived materials. However, by transitioning to a bioeconomy, there is the potential for improved supply security and a significant reduction in GHGs. Where leakage into the environment does occur, future plastic products should be designed such that their biodegradation does not have a negative impact on local, or global, ecosystems. At present, the biodegradation of biobased plastics is not guaranteed; bio-PE and bio-PET have exactly the same durability as their fossil-derived counterparts. Even plastics considered biodegradable, such as PLA, have vastly different decomposition rates, which are dependent on the specific material (for example, its thickness, formulation, crystallinity, and wettability) and the local environment. Industrial composting is generally much faster than home composting, while differences between terrestrial, freshwater, and seawater environments result in different biodegradation rates for both PLA and PHAs. However, when relevant standards are followed, these materials can eliminate the presence of micro- and nanoplastics, as the material is converted predominantly into CO_2_, H_2_O, and CH_4_. This carbon can then be captured by biomass (such as sugar cane) or used directly in gas fermentation processes in the manufacture of new biobased plastics. Therefore, the widespread adoption of these materials has the potential to efficiently recycle carbon and facilitate a zero-waste future.

## Figures and Tables

**Figure 1 polymers-16-01621-f001:**
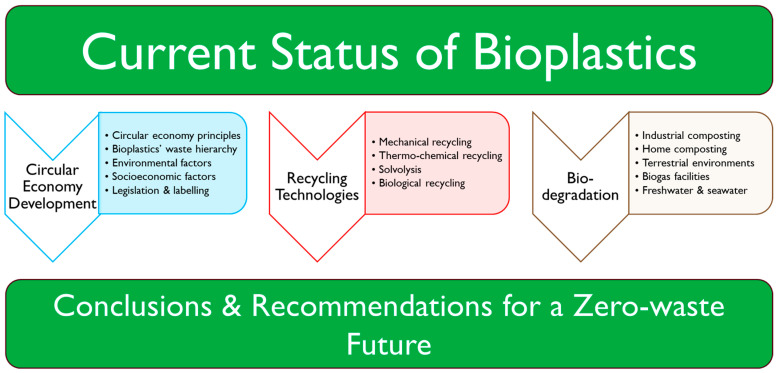
Workflow and concepts discussed in this article.

**Figure 2 polymers-16-01621-f002:**
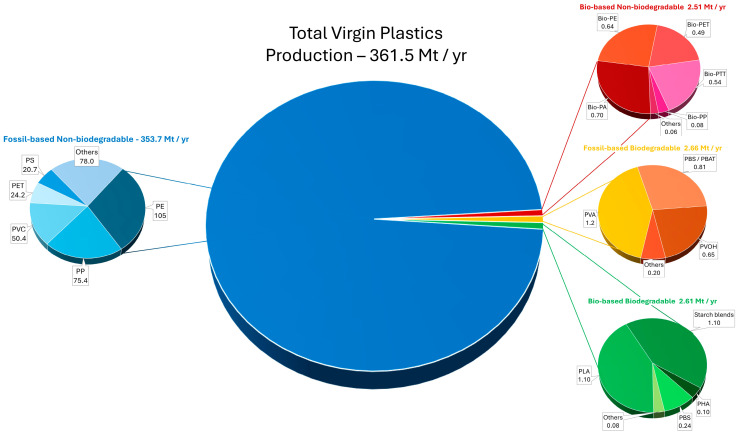
Annual production of different plastic types. All data in mega tonnes (Mt) and taken from Refs. [35,75,76,77]. Note that this does not include the ~30–40 Mt of recycled plastic which also enters the supply chain.

**Figure 3 polymers-16-01621-f003:**
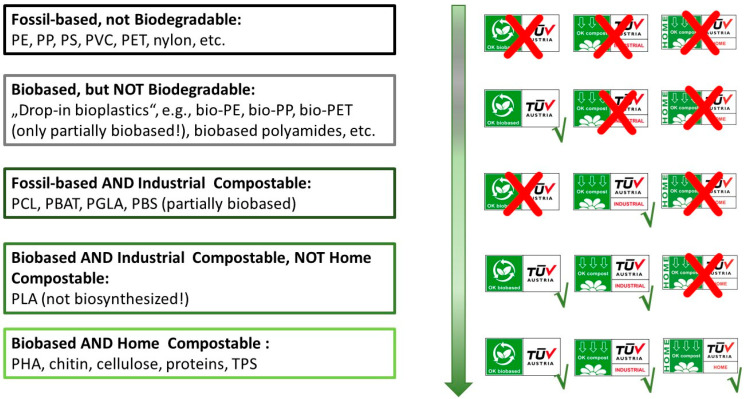
Different types of “bioplastics”, indicating the attributes that make them “bio”, according to the established TÜV Austria test scheme for “biobased”, “industrial compostable”, and “home compostable”.

**Figure 4 polymers-16-01621-f004:**
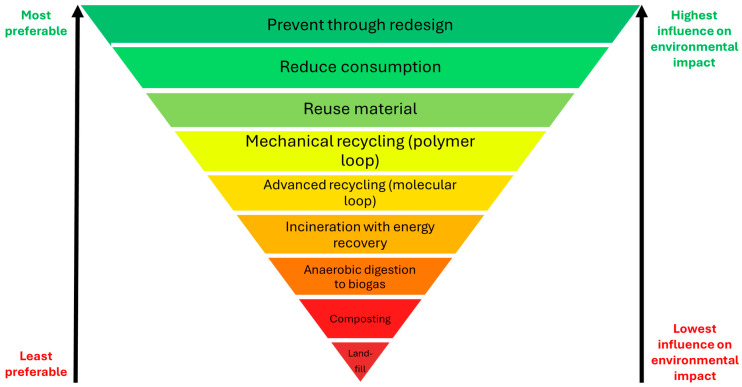
A waste hierarchy for biodegradable plastics.

**Figure 5 polymers-16-01621-f005:**
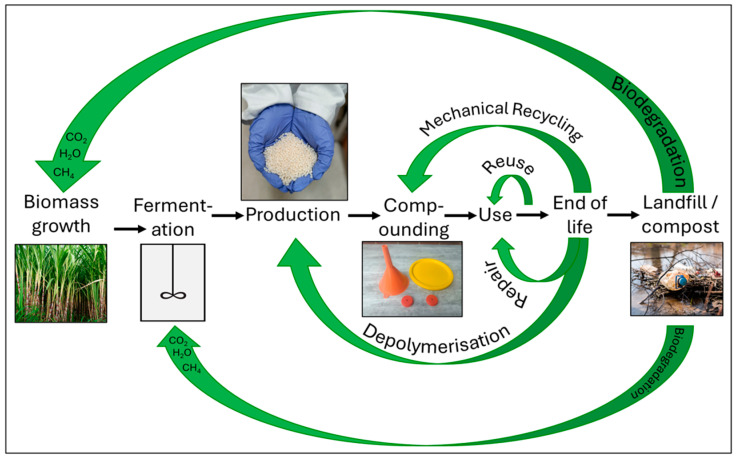
A circular economy for bio-sourced, biodegradable plastics.

**Figure 6 polymers-16-01621-f006:**
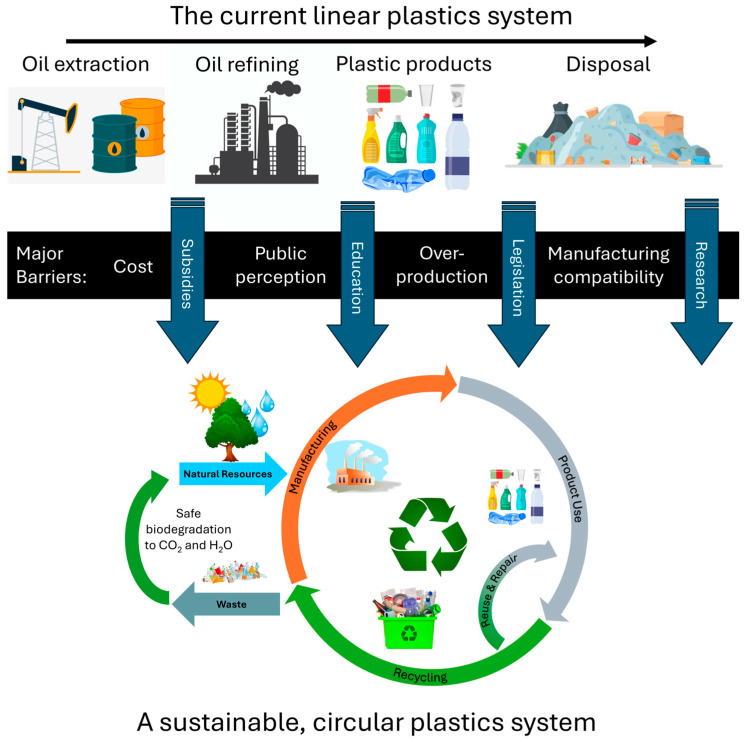
Barriers and enablers of transitioning from a linear to a circular plastics system.

**Table 1 polymers-16-01621-t001:** Summary of social issues considered within S-LCA (adapted from Ref. [120]).

Stakeholder Group	Social Issue
Workers	Freedom of association/collective bargaining Child labor Fair salary Working hours Forced labor Equal opportunities/discrimination Worker health and safety Social benefits
Consumers	Feedback mechanisms End-of-life responsibility Consumer health and safety Consumer privacy Transparency
Local community	Access to material and immaterial resources Delocalization and migration Cultural heritage Local employment Safe, secure, and healthy living conditions Respect of indigenous rights Community engagement
Society	Technology development Public commitments to sustainability Contribution to economic development Prevention and mitigation of armed conflicts Corruption
Value chain actors	Supplier relationships Fair competition Promotion of social responsibility Respect of intellectual property rights

## Data Availability

No new data were created or analyzed in this study. Data sharing is not applicable to this article.

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
