# Peer review of "Carbon Recycling of High Value Bioplastics: A Route to a Zero-Waste Future"

_polymers, 2024, doi:10.3390/polym16121621_

Round 1
Reviewer 1 Report
Comments and Suggestions for Authors
The manuscript entitled ‘Carbon recycling of high-value bioplastics: A route to a zero-waste future’ by Matthew Keith et al is an interesting work for publication. The present work is enlightened on bioplastic, with the theme of carbon recycling as a path toward achieving a zero-waste future through producing and utilizing high-value bioplastics. This work will be beneficial for environment mitigation readers. Nevertheless, the present structure and content of the scientific article in not sufficient to meet the standard of the journal. I, therefore, feel that this manuscript needs minor revision.
Comments:
1. The authors should use eye-catching and text-hidden keywords, so that reader’s curiosity can be attracted to the article.
2. The introduction requires more literature focusing on Bio-sourced and biodegradable polymers-based materials, with few recent works.
3. All images should be in high resolution with discussed critically. Moreover, work should be diagrammatic, it may be the author’s creative images or maybe from copyright permission.
4. Figures 1-3 should be interpreted comparatively and critically.
5. Moreover, the present scenario of bioplastic, management of nanomaterial wastes, industrial solid waste, industrial composting, home composting, biodegradation in terrestrial environments, biodegradation in seawater, etc should be present in a tabular format containing 25-30 entries. The authors are also recommended including the following similar works to enhance the literature-
Management of nanomaterial wastes/https://doi.org/10.1016/B978-0-323-90982-2.00007-X; Industrial solid waste: An overview/https://doi.org/10.1016/B978-0-323-85604-1.00018-4;
6. Reference style is not uniform, it should be checked. All references should be in a similar style, in some references the journal name is in abbreviation form while others are in full form.
7. The conclusion should be the output of the work/finding of the work. It should be in a single paragraph.

Minor editing of English language required.
Author Response
Thank you very much for reviewing this article and providing detailed comments. In response to these, we have provided the list below with each number mapping to your initial comment. All new text (as a result of suggests from yourself and the other reviewers) have been highlighted in yellow in the revised manuscript.
- Thank you very much for suggesting the use of more keywords. “biodegradation; recycling; waste management; circular economy; biopolymers” have all been added. It is not currently clear on the MDPI website how to include text-hidden keywords and guidance has been sought from the editor.
- This is a good point – the Introduction largely focusses on the current state of fossil based plastics. Some additional sources which focus specifically on bioplastics are now included in the final paragraph.
- Additional figures (Figure 1, Figure 2, Figure 6 in the revised manuscript) have now been updated with high res images provided during upload. They are critically discussed within the text with new sections highlighted in yellow.
- It is the authors’ belief that Figures 3 to 5 (previously 1 to 3) are used to illustrate the principles described in the text and so critically analysing these figures is not appropriate here. Instead, a reference has been made to the barriers and enablers which are now discussed in section 6. Some comparative description linking Figure 4 (waste hierarchy) to Figure 5 (a circular economy diagram) has also been added.
- Thank you for this suggestion. Table 2 with 42 entries has now been added to summarise biodegradation in home & industrial composting facilities, anaerobic digestion, terrestrial environments, seawater and freshwater. Thank you also for providing the additional two references which provide an excellent background into the issues of industrial solid waste and management of nanowaste. These have now been referred to in the introduction.
- Thank you for spotting the issue with the referencing – these have all been reviewed and corrected.
Thank you for your comment on the conclusion being in one paragraph. This has been reviewed and, as it is a lengthy article, there are multiple findings of this work, which we believe sit best within their own paragraph to maintain clarity. Therefore, we would like to keep the current structure of the conclusion, especially since this has not been highlighted as an issue by the other three reviewers.
Reviewer 2 Report
Comments and Suggestions for Authors
Authors have talked about a very significant topic in this manuscript, carbon recylcing of high value bioplastics. The wave and demand for sustainability is at its epitome in numerous polymer industries and this is a great addition to the knowledge pool. Following are my comments and suggestions.
1. References: The authors have done a very good in distinguishing the bioplastics/biodegradable by stating the standards. I would suggest citing very relevant review articles that talk about the biobased, microplastics, etc in detail as a holistic polymer sustainability challenge. Example: https://doi.org/10.3390/su152215758. Also suggested is to mention/cite review articles that talk in detail about the PHA biodegradability and its composites. Example: https://doi.org/10.1039/D0GC01647K . both of them will surely enable readers to refer to important analogous articles.
2. Though not exactly falling under the "microplastics" tire wear particles also contribute a lot to the plastic pollution, please mention the same as a possible source and cite articles that talk about it. It can be the tire particles present in roads, soil, air, water etc, with adverse effects.
3. The criteria mentioned for bioplastics, please use ISO/ASTM standards relevant to bio-based too in addition to the eco label section? Also do mention that some companies resort to biomass approach incorporating a "renewable feedstock/solar/renewable energy" in the feedstock production stage.
4. I understand you have concentrated on Europe eco labels and certifications, please also mention such in America region, that would be USDA's biobased label and certifications by mostly ISCC+ via ASTM 6866 standards via C14 isotype determination by advanced analytical technique.
Kudos on the nice piece of work!
Author Response
Thank you very much for reviewing this article and providing detailed comments. In response to these, we have provided the list below with each number mapping to your initial comment. All new text (as a result of suggests from yourself and the other reviewers) have been highlighted in yellow in the revised manuscript.
- Thank you also for the additional sources. Your first example has been referred to in the introduction to highlight the need that addressing plastic pollution needs to be done holistically. Your linked review article about PHA biodegradability has been highlighted in Section 5. Biodegradation along with referring to additional reviews summarising the biodegradability of PLA, TPS, and their composites.
- This is a very good point about tyre wear contributing towards microplastic pollution. This has now been considered in the introduction with 4 new references added.
- Reference to ISO, ASTM and EN standards has now been included in section 2, along with a link to the aforementioned biomass approach.
- Yes, as the authors are based in Europe, the focus is on European certification. However, including a reference to US certification schemes is a valuable addition, which has now been included in section 2.
Reviewer 3 Report
Comments and Suggestions for Authors
Carbon recycling of high value bioplastics: A route to a zero- waste future
Manuscript number: Polymers 3034717
The work covers comprehensively considered the areas related to the aspiration to create conditions for zero waste emission in the future. The authors have exceptionally analytically summarized the areas related to the current state of production of bioplastics, the principles of the circular economy related to the production of bioplastics, the hierarchy of bioplastic waste, socio-economic factors of bioplastic production, legislations, Recycling Technologies and biodegradation.
Some minor improvement is necessary regarding:
1.. Consideration of PET recycling.
2. More graphical, schematic and tabular presentation in supplementary materials.
No further review is necessary.
Author Response
Thank you very much for reviewing this article and providing detailed comments. PET recycling was already mentioned in section 4.1 and 4.3, however some additional details have been added in both of these sections. Based on the advice of the other reviewers, graphical and tabular presentations have also been added to the article (Figures 1, 2, and 6 and Table 2). All new content (except the figures and table 2) is highlighted in yellow for ease of identification.
Reviewer 4 Report
Comments and Suggestions for Authors
- Add more figures and tables to the manuscript and discuss their significance.
- Include an overall schematic diagram to illustrate the main workflow or concepts.
- Add a schematic diagram to depict future perspectives and challenges.
- In each section below, write the author’s view and conclusion.
Author Response
Thank you very much for reviewing this article and providing detailed comments. We have now included Figures 1, 2 and 6 and discussed their significance in the text. Figure 1 is an overall schematic to show the workflow of this paper, while Figure 2 illustrates the proportion of different plastics produced and Figure 6 summarises future perspectives, barriers and enablers to achieving a more sustainable plastics system. The authors’ views and conclusion are present in the article, if additional points are needed here, could you please clarify what is meant by adding views “in each section below”?